# THE SURPRISING BEHAVIOR
# OF CONVOLUTIONAL GRAPH NEURAL NETWORKS

## ABSTRACT

We highlight a lack of understanding of the behaviour of Convolutional Graph Neural Networks (GNNs) in various topological contexts. We present 4 experimental studies which counter-intuitively demonstrate that the performance of GNNs is weakly dependent on the topology, sensitive to structural noise and the modality (attributes or edges) of information, and degraded by strong coupling between nodal attributes and structure. We draw on the empirical results to recommend reporting of topological context in GNN evaluation and propose a simple (attribute-structure) decoupling method to improve GNN performance.

## 1 INTRODUCTION

Convolutional Graph Neural Networks (GNNs) have produced state-of-the-art results in areas which utilize graph data (Wu et al., 2019). Despite their widespread and rapid application across many fields, little research has been conducted on understanding the effect of topology on GNN behavior, especially in the context of attributed networks. Though GNNs use both graph modalities i.e., the topology and nodal attributes, it is not clear whether they utilize topology to the same degree as nodal attributes or if they generalize across topological contexts.

In this paper, we seek to underscore this lack of understanding. We present 4 empirical studies, which characterized and compared GNNs' utilization of topology, with surprisingly counter-intuitive results.

*Topology, Does it really matter?:* We empirically analyze GNN performance and show that it is, contrary to expectation, **only loosely dependent on the nodal topological characteristics of a graph** - particularly those of connectivity. The impact of connectivity is explored in the extreme case of a disconnected graph with multiple components (Section 3).

*Just Noisy Graphs:* We analyze the impact of topological perturbations on the models' performance and dependence on topology. Given the weak dependence of GNN performance on topological features and the neighborhood aggregation mechanisms of GNNs, one would **expect the graphs to be robust to topological noise.** However, we see that **GNN performance degrades considerably with noise** (Section 4).

*Attributes & Topology, together or not?:* We show that instead of improving performance (Fosdick & Hoff, 2015), **increased coupling between the modalities of topology and nodal attributes hampers it**. We demonstrate a simple method to decouple the topological and attribute information which improves performance by acting as a regularization mechanism (Section 5).

*Attributes vs. Topology:* We then create an experiment which questions whether the information in the graph structure and the attribute stack has any **overlap and is inter-convertible by simple means** (Section 6).

The counter-intuitive results of these experiments highlight a gap in understanding of the behavior of GNNs in various common scenarios (Du et al., 2017). This gap prohibits applying these powerful models to sensitive and diverse areas such as medicine (Parisot et al., 2018), chemistry (De Cao & Kipf, 2018), and governance (Li & Goldwasser, 2019). The consequent recommendations from this study take the first step in shrinking that gap.

## 2 METHODS

In this work we compared the behavior of a set of GNNs across engineered graphs derived from a basket of benchmark datasets.

**Evaluation:** The models were evaluated in a transductive node classification setting that closely follows the evaluation setup in Shchur et al. (2018). For each model, we used a fixed set of hyperparameters reported in Shchur et al. (2018), which are the best-performing configurations that have achieved the best average accuracy on *Cora* and *Citeseer* datasets (averaged over 100 train/validation/test splits and 20 random initializations, using only the largest-connected-component. In reporting test accuracies, unless otherwise mentioned, we report the accuracy averaged over 20 train/validation/test splits and 2 random initializations for each model.

**Models**: We consider the most prolific spectral and spatial GNNs, and compare against an attention-based GNN. In this work, we study Graph Convolutional Network (GCN) (Kipf & Welling, 2016), GraphSAGE (GS) (Hamilton et al., 2017) and Mixture Model Network (MoNet) (Monti et al., 2017), and Graph Attention Network (GAT) (Veličković et al., 2017) from each category respectively. We also considered two additional non-GNN based methods: Label Propagation (LabelProp) and Label Propagation with Normalized Laplacian (LabelProp NL). These methods only consider the graph structure and not its node attributes.

**Datasets**: We use the following Citation, Co-author and Co-purchase networks (Table 2). Each network consists of an edge set and nodes that form the underlying topology. Attached to each node is an attribute stack (or feature vector), which describes metadata associated with the node, and a label, which is the target of a node classification algorithm. Citation networks, used to evaluate GCNs and GATS: *Cora*, *Citeseer* (Sen et al., 2008) and *Pubmed* (Namata et al., 2012), have documents as nodes, citation as edges and bag-of-words from the papers as node features. We consider a Co-author network, *Coauthor Physics*, with authors as nodes, paper co-authorship as edges, and keywords of the author's papers as node features (Shchur et al., 2018). We also consider Co-purchase networks: *Amazon Computers* and *Amazon Photo* (Shchur et al., 2018), with goods as nodes, co-purchase as edges and product reviews as node features.

## 3 TOPOLOGY, DOES IT REALLY MATTER?

While convolutional neural networks work with highly regular neighborhood structures (4-neighboring pixels), GNNs attempt to deal with situations where such regularity assumptions may not apply. They characterize nodes through the composition of their neighborhood either explicitly, as in spatial methods, or implicitly, as in approximate spectral methods. Thus, a denser neighborhood - one with more samples - would allow a more certain characterization of a node. In the following experiment, we investigate whether such an intuitive relationship with the underlying topological features, which proxy topological connectivity, exists.

**Question:** Do topological features impact GNN performance? In other words are nodal topological features good predictors of GNN performance?

**Hypothesis/Expectation:** Better connected nodes have larger neighborhoods and so should show less variance in GNN performance. Nodal topological characteristics should significantly impact performance.

**Counter-intuitive results:** Nodal topology does not appear to strongly impact the performance of the GNN, even in the extreme case it appears to degrade performance only after a critical point.

**Methods:** We selected features from Table 1 that can be defined at a node granularity. For each node, we computed these features and analyzed their correlations with the average classification accuracy for that node across test-train and initialization splits. Subsequently, we confirmed these results through a robust binning mechanism and a Mann-Whitney-U (MWU) test.

We then analyzed the extreme case of completely disconnected components, and measured the impact of increasing number and size of disconnected components in the graph. For each dataset, we considered subgraphs with different types and number of components being retained as the input graph to a model: (1) only the largest connected component, (2) all components except isolated nodes, and (3) all components.

**Results:** Surprisingly, the classification accuracy of node does not appear to depend strongly on its topological characteristics. As Fig. 1 shows, all topological features considered showed weak correlation ($|r| < 0.2$) across both models and datasets with significance ($p < 0.05$), with most indicators of connectivity being mildly positive. Though all the datasets considered are relatively

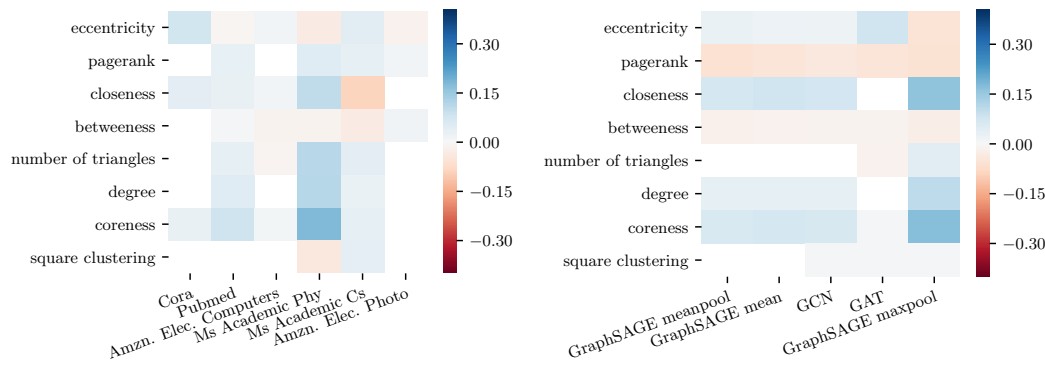

(a) correlation across datasets          (b) correlation across models

Figure 1: Accuracy and topological features show weak correlation (Pearson's $r$, significance $p < 0.05$) across **a**: Datasets **b**: Models. Finer granularity (per dataset per model) versions are in Appendix C

sparse ($D < 10^{-3}$), (*Amazon Photograph and Computers*) show slightly better connectivity properties through long-tailed degree and coreness distributions. This increased connectivity did not affect the performance. These are supported by a visual analysis and MWU test which shows significance ($p < 0.001$) in *Coauthor Physics, Pubmed,* and *Coauthor Computers* networks. In concordance with our intuition models that showed a positive mean-accuracy correlation with coreness, closeness and eccentricity also displayed moderate negative correlation ($0.2 < |r| < 0.4$, $p < 0.05$) with the $\sigma$ of the accuracy.

We then drilled into the effect of extreme disconnectvity in the form of disconnected components. We chose 4 graphs that represent scenarios which represented a range multi-component properties, as seen in Table 2. In all scenarios, GNNs (except GCNs) are able to handle graphs with multiple components better than Label Propagation baselines. We also observe that GNNs find multiple-component scenario presented in *Citeseer* the hardest to cope with; on average, GNNs perform 20% worse with the multi-component *Citeseer*.

For the other three datasets, it may seem that the overall model accuracies change negligibly despite the addition of multiple components. However, the reported model accuracies are an average of test nodes' performance; since most of the randomly selected test nodes are likely from the largest component, nodes located on the largest component would mask the poor node-level performance of other nodes. This is supported by Figure 2b, where we observe that nodes located in components other than the largest have a significantly worse performance than those located in the largest component. The sole exception is the *Amazon comp.* dataset, which shows a non-monotic irregular behavior amongst the smaller sized components. For GNNs across datasets, the average node accuracy on the largest component is 26.0% higher than those on other components; for LabelProp, this is even worse (with a 51.6% difference).

Finally, given the discrepancies between the performance of GCNs and other models in handling multi-component graphs, we further analyzed the impact of the number of components in the graph on the performance of GCNs and GATs. In this experiment, for each graph, we input the subgraph retaining only the largest $n_c$ components, with $n_c \in \{1, 2, 10, 100, N_c - N_i, N_c\}$, $N_c$ as total number of components and $N_i$ as number of isolated nodes. Figure 2a reports the test accuracies achieved at these intervals. We find that as $n_c$ increases from 1 to $N_c - N_i$, there is only a slight decrease in accuracies suffered by GCNs and GATs which are within the statistical limit. Again, we see that there is only a small divergence in performance between GCNs and other GNNs as the graph includes isolated nodes. GATs, as well as other GCNNs, cope well with the presence of isolated nodes. A comprehensive comparison of the effect of isolated nodes can be seen in Table 5.

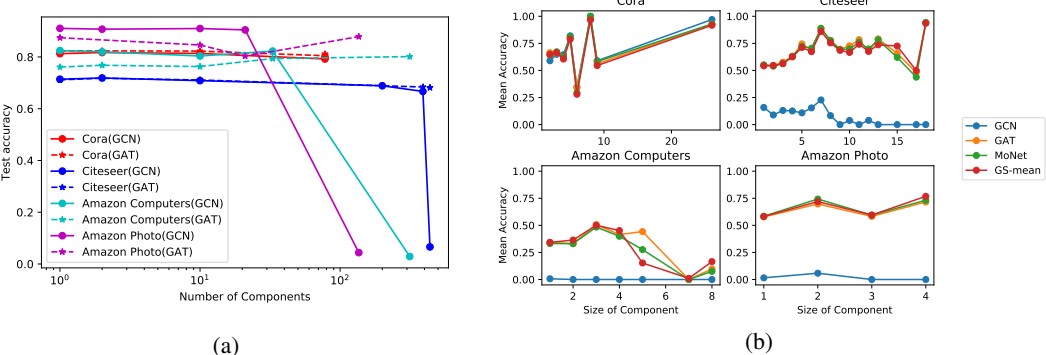

(a)                               (b)

Figure 2: **a**: Test accuracy for GCNs and GATs for graphs with varying number of components retained. Both models are robust upto a critical breakdown limit. **b**: Mean test accuracy per component against its size. Only components other than the largest component are included in the plot. Similar patterns are shown by all GraphSAGE models so only GraphSAGE-mean is shown. Larger components show better performance in all datasets with the exception of Amazon computers.

# 4   JUST NOISY GRAPHS

Structural errors may originate during the acquisition of networks (e.g. a malfunction in a cellular network) or during the construction of networks from underlying data, particularly in dynamic networks where transient phenomenon are difficult to separate from noise. Thus, understanding the robustness of GNNs to errors of a topological nature is essential to applying them successfully in real world situations. Having established that GNNs do not strongly depend on topological features in Section 3, it is reasonable to hypothesize that they should be robust, to all but specific, topological noise (Zügner et al., 2018). We restrict our investigation to edge-based perturbations and measure the performance of GNNs on introducing specific controlled perturbations to the structure, allowing us to characterize the behavior of GNNs in noisy scenarios.

**Question:** How does the addition of topological noise impact GNN performance?

**Hypothesis/Expectation:** As GNNs do not seem to depend on the topological features, they should remain fairly robust to noise and should continue to remain apathetic to topological features even in noisy circumstances.

**Counter-intuitive results:** The addition of noise degrades performance significantly in some models and even more surprisingly, the GNN performance becomes more dependent on the global topological characteristics in noisy graphs.

**Methods:** We constructed several perturbed versions of the *Cora*, *Citeseer*, *Pubmed* and *Coauthor Physics* graph datasets for this experiment - these span and represent different categories of connectivity and annotation (attribute stack) of networks. *Cora* has no isolated nodes and is one the best connected graphs (Table 1). *Citeseer* contrasts sharply in that it has the largest number of components.*Pubmed* and *Coauthor Physics* have the smallest and largest attribute stacks respectively. The perturbation method is derived from an Erdos-Renyi graph-based noise model, which can be shown to form the equivalent Gaussian noise in graphs. This allows use the equivalent of the central limit theorem in graphs (Miettinen et al., 2018). The details of the algorithm and parameters can be seen in Appendix E. It perturbs an edge with probability $\mathcal{P}$, and thresholds the resulting edges with respect to a parameter $t$. We picked a low $\mathcal{P} = 0.001$ to stay true to the original graph and considered 5 perturbed versions using $t \in \{0.3, 0.45, 0.6, 0.75, 0.9\}$ in our analysis.
We further analyzed the behavior of GNNs in the case of smaller perturbations ($t \geq 0.6$). As the perturbations are global rather than local, we used correlation analysis on the statistical characteristic of the global performance across graphs.

**Results:** The noise model produced graph with comparable topological characteristics (Table 4) which diverged from the original with increasing noise. Despite the topological similarity the addition of edge based noise degrades performance (Figure 3a). All GNNs handled noise better than LabelProp

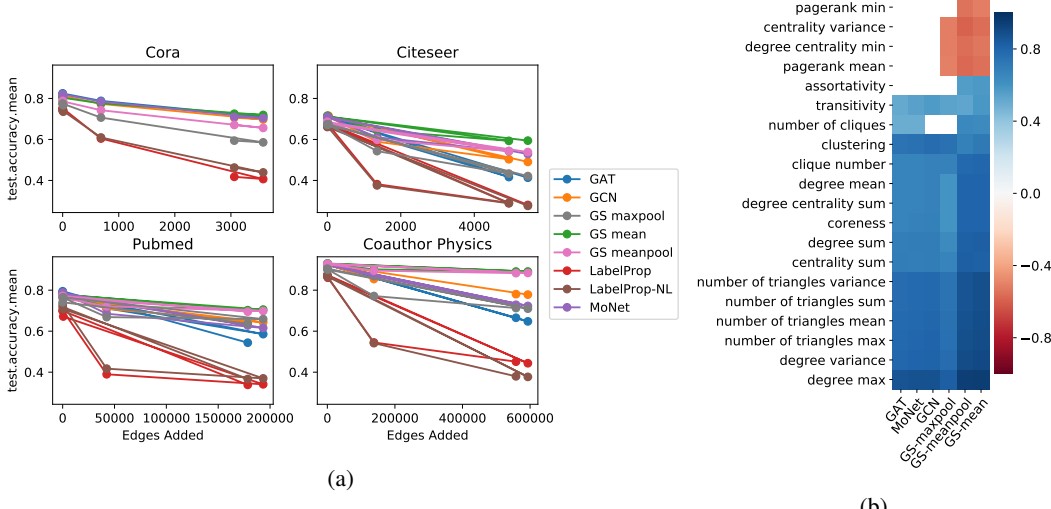

(a)

(b)

Figure 3: **a**: Mean accuracy of perturbed graphs with varying number of edges added. Lower thresholds imply more edges and more noise. **b**: Correlation between test accuracy and statistics of topological features for each model; only correlations with $r \geq 0.40$ and $p < 0.05$ are shown. Notice how the dependence on global statistics of topological characteristics increases drastically.

methods. However all models, even GNNs, degrade considerably as the number of edges added exceeds its original edge count; an exception is the performance of GraphSAGE (mean and meanpool) model on *Coauthor Physics*, which only saw an accuracy drop of $0.0006$ at $t = 0.3$, where more than $550,000$ edges had been added (i.e. adding about $2\times$ the original number of edges). This tolerance to the addition of edges might be related to the high sparsity of the original *Coauthor Physics* graph, which has an edge density of $0.0005$.

To analyze the how the topology effected GNNs in noisy environments. We considered the original and only high threshold (least noise) perturbed versions ($t \in \{0.6, 0.75, 0.9\}$) of 4 graph datasets. For each graph, we used the largest connected component as input graph to all GNN and LabelProp models. Figure 3b summarizes the significant correlations found between graph topological features and the average test accuracy achieved across all datasets per model. In stark contrast to the original graphs, there is a significant and strong positive correlation (mean $r = 0.74, p = 0.006$) with features which indicate better connectivity such as number of triangles , clustering (max $r = 0.766$), transitivity (max $r = 0.578$), coreness (max $r = 0.804$), and degree. There is also a significant negative correlation (mean $r = -0.54, p = 0.03$) with the minimum values of connectivity metrics. Correlations between these topological features with the standard deviation of accuracy across different train/validation/test splits also indicate that a higher transitivity ($r = -0.25$) and clustering ($r = -0.22$) correspond to a more consistent accuracy across splits.

This clearly indicates that, in the presence of noise, the performance of GNNs is more related to the topological features as compared to baseline LabelProp methods where no significant correlations are found. The uniform polarity of the correlations suggests that these correlations are consistent across different models and are related to the same changes in performance (albeit to varying magnitudes). Similar correlation patterns for validation $F_1$-scores support this conclusion.

## 5 ATTRIBUTES & TOPOLOGY, TOGETHER OR NOT?

GNNs use two modalities (topology and attributes) that are often coupled to characterize nodes e.g. high degree nodes in transport networks often correspond to interchange stations and have more amenities. A strong coupling has significant effects on the model's confidence and robustness (in the previous example, one can look at either the amenities present at the station or its degree to establish if it is an interchange) (Fosdick & Hoff, 2015). However, statistically establishing its impact is difficult due to the lack of graphs within the same domain with varying degrees of coupling. To overcome

this, we created variations of the graph that removed the coupling and statistically compared them to the original.

**Question:** Does the degree of association (coupling) between a node's topology and its attributes affect a GNN's performance?

**Hypothesis/Expectation:** Decoupling the modalities of topology and attributes, by shuffling nodal attributes would degrade performance.

**Counter-intuitive results:** Shuffling attributes while preserving nodal labels resulted in improved performance, whereas shuffling attributes while preserving nodal labels degrees added little improvement.

**Methods:** To selectively study the effect of attribute-topology coupling, we permuted (shuffled) the attribute vectors while keeping the underlying network topology constant. The shuffling mechanism decoupled the attribute vector from the nodal topology, while ensuring the attributes themselves were representative of the domain. Thus generating domain-faithful samples which facilitated a statistical analysis of the effect of the coupling. The attribute vectors were shuffled subject to certain constraints which impose a partition on the graph's nodes:

1. Shuffled without restriction (Naive).
2. Shuffled within partitions formed by class labels (Iso-Class). This ensures that the correspondence between both the attributes and class labels and topology and class labels is maintained.
3. Shuffled within partitions formed by class labels and node degree (Iso-Class-Deg). This additionally preserves correspondences between the node degree and class labels.

The original un-shuffled graph was compared with 10 attribute-shuffled variations for each dataset. The results were subsequently analyzed for significance using a grouped MWU test.

**Results:** We analyzed 4 datasets with 4 models (Figure 4). As expected, Naive shuffling led to a large decrease in performance across all models. However, surprisingly, shuffling the attribute vectors within Class-partitions led to an increase in mean accuracy. Statistically significant (MWU $p < 0.001$) mean increases ranging from $2.7\%$ to $6.8\%$ were seen across datasets for each GNN with MLPs being agnostic to the shuffle. GCN saw the largest increase (accuracy $6.8\%$ and $F_1$ $6.5\%$, MWU $p \sim e - 20$)

Even more counter-intuitively the performance remained same, and even dropped slightly but not significantly, when partitions were redefined using both the degree and Class labels. Similar results ($2.4 - 6.5\%$ MWU $p \sim e - 20$) were seen for the $F_1$ score across all GNN models and all datasets. MLP, which does not take topology into account, showed no significant change on shuffling.

Therefore, selectively decoupling attributes from the graph structure by constricted shuffling leads to a consistent and significant improvement in accuracy without corresponding loss in $F_1$. Implementing it is simple and has the potential to improve performance across models.

## 6 ATTRIBUTES VS. TOPOLOGY

Edges can encode information between nodal attributes that may not themselves annotate the Network. For example, in a shipping network between cities, supply and demand are nodal attributes that often correlate across edges (shipping routes). When using graph inference or edge prediction, we take a set of points with attribute vectors (and potentially incomplete sets of edges) and use them to infer the edges that should exist in a graph. Constructing graphs via this method is common in areas such as protein or transcription networks.

Often, there is some overlap in the information used to create edges and the attributes annotating the nodes. This begs the question, are these two modalities equivalent and can one be converted into the other?" In the previous sections, we have demonstrated that GNNs treat the graph structure (topology) differently, and are sensitive to perturbations to it. If information could be encoded in either modality, because they are equivalent, it would allow for the construction of more robust networks. These experiments therefore explore this equivalency to determine whether it is possible to encode attribute information into the graph structure.

**Question:** Is it possible to convert attribute data to structural information?

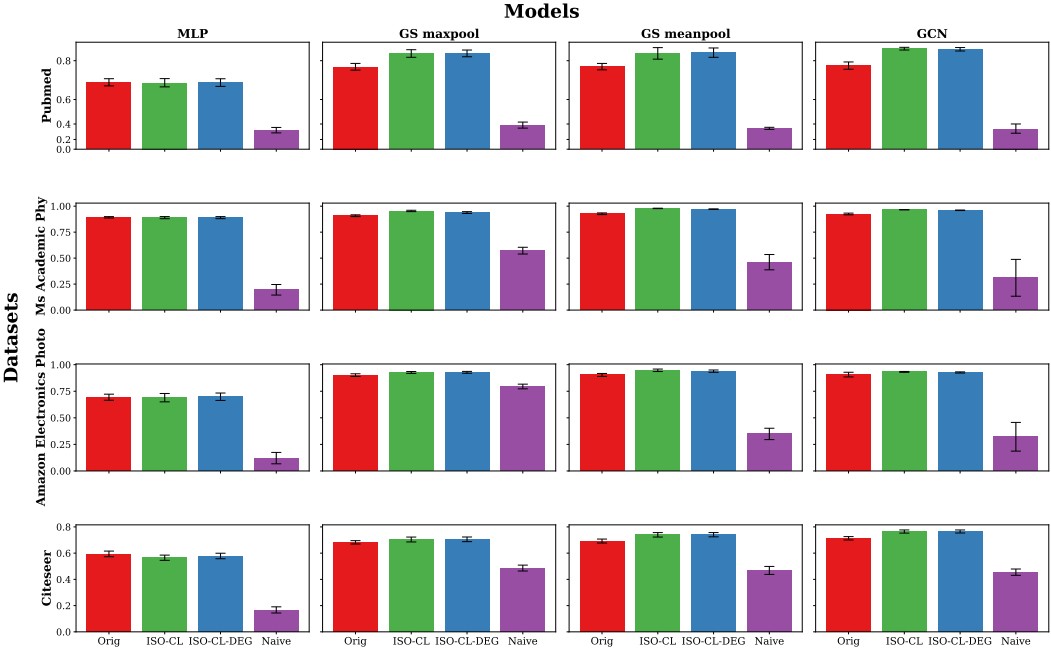

Figure 4: Accuracy for different shuffle scenarios. We see that shuffling within partitions imposed by class labels (ISO-CL and ISO-CL-DEG) leads to better results. The grid-y-axis indicates different datasets and the grid-x-axis different model types. The x-axis shows different constraint scenarios and the y-axis is accuracy scaled logarithmically. The error bars indicate standard deviation across runs.

**Hypothesis/Expectation:**

1. Decorrelating the nodal attributes should lead to better or constant performance at worst.
2. Increasing dimensions discarded from the nodal attribute stack would correspond to a decrease in performance.
3. Retaining higher variance dimensions should yield better performance.
4. Adding edges in place of lost dimensions should alleviate the decrease in performance.

**Counter-intuitive results:** PCA decreases performance. Augmenting the edgeset with edges that ought to be there does not alleviate the situation.

**Methods:** We constructed graphs such that a portion of the attribute information was embedded in their edge set. We chose three datasets that represent attribute stacks of different dimensions with *Pubmed* having the smallest (500 attributes), *Amazon Photo* a middling (745 attributes), and *Citeseer* having the largest (3703 attributes).

Rather than randomly selecting which attributes to discard, which would be computationally prohibitive for large attribute sizes, the attribute set was transformed using a PCA transform and the $k\%$ highest variance components are retained while the remainder are discarded. The discarded attributes are then either used to augment the edgeset or discarded out of hand. If they are used to augment the edgeset, the cosine similarity, between the attributes to be discarded is measured and the most similar pairs identified. An edge for each pair identified is placed in the edgeset. The two scenarios are contrasted statistical differences measured using and MWU test. Separate graphs for ($k \in \{99, 97, 94, 92, 90, 85, 80, 75\}$) were constructed to analyze the effect of number of attributes discarded.

We also contrasted this to the opposite scenario where the $k\%$ lowest variance components are retained and the remainder used to augment the dataset. Fig. 7 details the 4 types of graph constructed and the difference in their edge sets and attributes stacks. We use cosine similarity due to the nature of the attributes which are either bag-of-word, binary, TF-IDF vectors. The normalized cosine measure allows for a more semantically accurate comparison between vectors. Despite the careful selection

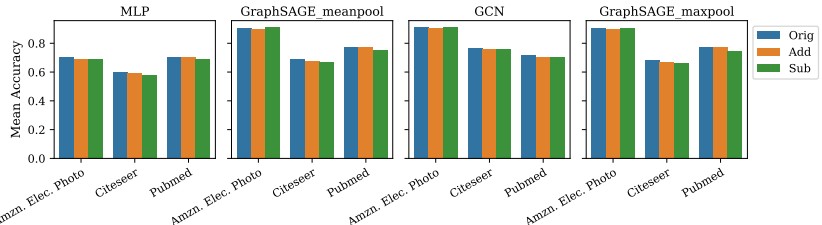

Figure 5: The mean accuracy in when all the attributes are retained but the edge set is augmented or reduced. Edges identified by high correlation between attribute vectors are added or removed from the edge set. We see a clear decrease in performance on adding correlation. Subtracting common edges removes additional redundancy and decreases performance further.

of an appropriate metric, there may still be noisy pairs. To alleviate this we only selected the most similar pairs and tried to keep the number of added edges of the same order as the parent graph.

Differences in performance across 3 initializations and 15 test-train splits were verified for significance through MWU tests.

We drilled down into edge case where all attributes were retained with a different set of constructions. Here, the goal was to analyze redundancy of information between modalities more explicitly. We compared GNN performance with pairs of graphs constructed such that the edges between the most similar nodes were used to either augment the edge set or diminish it (set difference).

**Results:** The PCA version of the attribute set elicited an average $50\%$ performance drop across all models, even with $> 99\%$ of the features retained. Increased reduction of the features did not yield a monotonic decrease ($|\rho| < 0.1$; as the number of points were limited we also verified the monotonicity visually). There was little difference between cohorts retaining high variance and low variance attributes. The performance was also insensitive to the addition of the edges based on the discarded features.

In the case where all attributes are retained, there is a relatively small overlap ($< 10\%$) between the edge set created through similarity and the original edges. While adding extraneous edges degrades performance, removing common edges degrades it further, albeit by an insignificant amount.

The addition of edges to the graph is associated with a stronger (negative) dependence on the topological features of individual nodes. The models show strong ($|\rho| > 0.3$) statistically significant ($p < 0.05$) and consistent correlation. Betweenness was one topological characteristic which showed little correlation even on the addition of edges. We also robustly compared the set of nodes which tend to be misclassified (accuracy $< 0.4$) to those that tend to be accurately (accuracy $> 0.6$) classified using the MWU test. The test showed a significant (perhaps nonlinear) difference in the coreness, closeness and degree characteristics of the two node populations. Confidence in differences was higher in cases where a higher number of attributes were retained.

To conclude, it is clear that attribute information attributes themselves are sensitive to their representation and cannot be simply transformed into the graph structure.

## 7    DISCUSSION

As the experiment in Section 6 demonstrates, topology and nodal attributes are distinct modalities that do not easily convert from one to the other. This lack of interchangeability may be due to the inherently different descriptive capacities (an edge in an undirected, unweighted graph is only a binary value whereas many attributes have significantly larger domains), and their differential utilization by GNNs (network structure is primarily used to define a neighborhood). While the experiment is restricted by the simplicity and selection of the distance function/edge prediction mechanism, choosing a more sophisticated method would asymptotically be equivalent to solving the original problem (presented to GNN) itself. Future work could entail exploring possible distance metrics and their effectiveness in the pre-processing pipeline. It is also possible that a more elaborate hyperparameter tuning may yield better results. Future work to characterize the complexity of GNN

inference and training with respect to the topology and attribute stack would be important. This would facilitate the selective deployment of models to situations they are best suited for.

Given this substantial difference between the modalities, the restricted decoupling (explored in Section 5) removes any bias which would have been amplified by their association. Shuffling exposes the model to a larger number combination of neighborhoods and attribute vectors stack. It thus acts as a regularization mechanism and improves generalization of the models. Such a shuffling step would positively increase standard performance across a range of models and should be included as a standard component of GNN deployment pipelines.

The difference between modalities and their utilization does not fully explain why the weak dependence on topology in Section 3 increases across experiments on the addition of edges. One theory which may explain the empirical results presented here originates from the analysis of the nodes influencing (nodal domains) a graph's eigenvectors in sparse situations provides additional insight. An eigenvector induces partitions on a graph's nodes based on the sign ($\pm$) of its (significant) components. Each maximally connected subgraph with a similar sign is called a nodal domain. With increasing sparsity (worse connectivity (Albert & Barabási, 2002)), the nodal domains of the eigenvectors tend to de-localize and increase in number (Arora & Bhaskara, 2011). This implies that the nodes represented by the eigenvector are further away and more fragmented. Thus making the complete characterization of a node dependent on a larger neighborhood and needing more samples to control its variance. The relatively sparse benchmark datasets used for our experiments would require more than the immediate neighborhoods, considered by most GNNs, to characterize each node. This is turn would force the GNN to pay less attention to the topology. In noisy, but dense, situations (Section 4), the GNN would pay more attention to the topology and as a result, would imbibe its error. While these studies would benefit from exploring alternative noise (e.g. block) models and a wider range of datasets, their findings clearly indicate that a thorough evaluation of a model must include the dataset's topological and noise features - features which preprocessing often changed unrealistically (e.g. discarding all but the largest component).

## 8 CONCLUSION

The use of GNNs across a variety of applications is growing rapidly. However, our poor understanding of their behaviour could seriously hinder our ability to fully utilise their potential. Our experiments demarcate an important gap in our understanding, and that an intuitive understanding of GNN behaviors is inaccurate. Our results show that although GNN performance is only weakly (although positively) correlated with a graph's topology, this changes in a number of common scenarios relevant to their everyday application.

They also show that GNNs are sensitive to structural noise. Therefore, to ensure a more thorough and complete benchmarking, we recommend that future models should consider and report topological and noise characteristics of datasets in the evaluation process. We also empirically evidence the limited inter-convertibility of attribute and topological information in a graph. Furthering this dissociation between modalities, we demonstrate an effective method of regularization through attribute shuffling. This study serves as a timely and important first step towards recognizing the need to address the network topological context when operating GNNs within the graph domain.

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

# Appendices

## A    TOPOLOGICAL FEATURES

Table 1: Graph topological features.

| Feature | explanation |
|---|---|
| size and order | Number of edges and nodes |
| degree | Statistical characteristics of the distribution of degrees (across nodes) in a graph |
| assortativity | A measure of mixing in graphs instantiated; by the correlations between node degrees |
| transitivity | Fraction of all possible triangles present in a graph |
| coreness | $k$-core's are maximal subgraphs where each node has degree $k$. Averaged across max. $k$ $\forall$ nodes |
| number of triangles | For each node total number of complete 3 node $K_3$ subgraphs that node is a part of |
| number of cliques | Total number of complete subgraphs |
| clustering | Clustering of a node measures the completeness of its neighbourhood; averaged across nodes |
| centrality and degree centrality | A (eigenvetor and degree respectively based) measure of node importance |
| communicability | For each node it is the sum of closed walks of all lengths starting and ending at that node |
| density | Fraction of actual to all possible edges |
| diameter and radius | Maximum and minimum eccentricity present in the graph |
| pagerank | Weighted ranking of the nodes in the graph based on the structure of the incoming links |

## B  DATASET CHARACTERISTICS

Table 2: Dataset statistics and description

| Dataset | Data | | | | | Component | | | |
|---|---|---|---|---|---|---|---|---|---|
| | Nodes | edges | Density | Features/Attributes | Classes | $N_c$ | $N_i$ | Largest | 2nd Largest |
| Cora | 2708 | 5278 | 0.00072 | 1433 | 7 | 78 | 0 | 2485 | 26 |
| Citeseer | 3312 | 4660 | 0.000424949 | 3703 | 6 | 438 | 48 | 2110 | 18 |
| Amazon Comp. | 13752 | 245861 | 0.001300138 | 767 | 10 | 314 | 281 | 13381 | 8 |
| Amazon Photo | 7650 | 119082 | 0.002035073 | 745 | 8 | 136 | 115 | 7487 | 4 |
| Pubmed | 19717 | 44324 | 0.00011402 | 500 | 3 | na | na | na | na |
| Coauthor Physics | 34493 | 247962 | 0.000208418 | 8415 | 5 | na | na | na | na |

Table 3: Topological features for each dataset

| Dataset | Topological | | | | | | | | | | | |
|---|---|---|---|---|---|---|---|---|---|---|---|---|
| | pagerank | | closeness | | betweeness | | number of triangles | | degree | | coreness | |
| | mean | std | mean | std | mean | std | mean | std | mean | std | mean | std |
| Cora | 3.7e-4 | 3.7e-4 | 1.4e-1 | 4.7e-2 | 1.7e-3 | 6.9e-3 | 1.8e+0 | 4.7e+0 | 3.9e+0 | 5.2e+0 | 2.3e+0 | 8.8e-1 |
| Citeseer | 3.0e-4 | 2.0e-4 | 4.5e-2 | 3.5e-2 | 1.0e-3 | 3.8e-3 | 1.1e+0 | 3.8e+0 | 2.8e+0 | 3.4e+0 | 1.7e+0 | 1.0e+0 |
| Amazon Comp. | 7.3e-5 | 1.0e-4 | 2.9e-1 | 6.2e-2 | 1.6e-4 | 1.9e-3 | 3.3e+2 | 1.6e+3 | 3.6e+1 | 7.0e+1 | 1.9e+1 | 1.4e+1 |
| Amazon Photo | 1.3e-4 | 1.4e-4 | 2.4e-1 | 5.1e-2 | 3.8e-4 | 2.2e-3 | 2.8e+2 | 9.0e+2 | 3.1e+1 | 4.7e+1 | 1.7e+1 | 1.1e+1 |
| Pubmed | 5.1e-5 | 6.4e-5 | 1.6e-1 | 2.0e-2 | 2.7e-4 | 1.6e-3 | 1.9e+0 | 8.4e+0 | 4.5e+0 | 7.4e+0 | 2.4e+0 | 1.9e+0 |
| Coauthor Physics | 2.9e-5 | 1.8e-5 | 2.0e-1 | 2.4e-2 | 1.2e-4 | 3.3e-4 | 4.1e+1 | 7.9e+1 | 1.4e+1 | 1.6e+1 | 7.7e+0 | 4.3e+0 |

Table 4: Mean Topological characteristics for the perturbed datasets $(t > 0.6)$ note that a threshold of 1 represents the original dataset

| origDataset | threshold | assortativity | num triangles | mean degree | clustering | clique number | pagerank | coreness |
|---|---|---|---|---|---|---|---|---|
| citeseer | 0.30 | 0.068450 | 1.076540 | 6.108696 | 0.053845 | 6.0 | 0.000302 | 3.103563 |
| | 0.45 | 0.070607 | 1.081975 | 5.796498 | 0.044370 | 6.0 | 0.000302 | 3.175574 |
| | 0.60 | 0.058629 | 1.057367 | 3.629227 | 0.103743 | 6.0 | 0.000302 | 2.026771 |
| | 0.75 | 0.049872 | 1.054952 | 2.830314 | 0.141505 | 6.0 | 0.000302 | 1.737017 |
| | 0.90 | 0.049428 | 1.054348 | 2.812198 | 0.142042 | 6.0 | 0.000302 | 1.733243 |
| | 1.00 | 0.049337 | 1.056159 | 2.814010 | 0.142555 | 6.0 | 0.000302 | 1.735205 |
| cora | 0.30 | -0.037038 | 1.874446 | 6.538405 | 0.088205 | 5.0 | 0.000369 | 3.706425 |
| | 0.45 | -0.043658 | 1.842097 | 6.158050 | 0.118800 | 5.0 | 0.000369 | 3.299778 |
| | 0.60 | -0.057992 | 1.811854 | 4.402511 | 0.195066 | 5.0 | 0.000369 | 2.515694 |
| | 0.75 | -0.065665 | 1.805207 | 3.905465 | 0.239960 | 5.0 | 0.000369 | 2.319239 |
| | 0.90 | -0.065821 | 1.805207 | 3.898080 | 0.240760 | 5.0 | 0.000369 | 2.315547 |
| | 1.00 | -0.065871 | 1.805761 | 3.898080 | 0.240673 | 5.0 | 0.000369 | 2.317208 |
| Coauthor Phy | 0.30 | 0.202883 | 42.606732 | 48.771983 | 0.024354 | 12.0 | 0.000029 | 31.750471 |
| | 0.45 | 0.206186 | 42.387644 | 46.734468 | 0.026188 | 12.0 | 0.000029 | 29.791697 |
| | 0.60 | 0.231953 | 40.806801 | 22.355782 | 0.108013 | 12.0 | 0.000029 | 11.729916 |
| | 0.75 | 0.201831 | 40.615284 | 14.451628 | 0.368153 | 12.0 | 0.000029 | 7.749254 |
| | 0.90 | 0.201097 | 40.613893 | 14.362045 | 0.377210 | 12.0 | 0.000029 | 7.708463 |
| | 1.00 | 0.201031 | 40.751747 | 14.377526 | 0.377623 | 12.0 | 0.000029 | 7.715507 |
| pubmed | 0.30 | 0.104907 | 2.321246 | 24.106609 | 0.004180 | 8.0 | 0.000051 | 15.736522 |
| | 0.45 | 0.107946 | 2.257646 | 22.597251 | 0.004447 | 8.0 | 0.000051 | 14.658264 |
| | 0.60 | 0.091103 | 1.930517 | 8.797180 | 0.015779 | 8.0 | 0.000051 | 4.764061 |
| | 0.75 | -0.039705 | 1.899173 | 4.549475 | 0.058748 | 8.0 | 0.000051 | 2.429173 |
| | 0.90 | -0.044067 | 1.898869 | 4.491353 | 0.060057 | 8.0 | 0.000051 | 2.392758 |
| | 1.00 | -0.043640 | 1.904955 | 4.496019 | 0.060175 | 8.0 | 0.000051 | 2.394533 |

## C    ACCURACY AND TOPOLOGICAL FEATURES

## D    MULTI COMPONENT PERFORMANCE

Table 5: Mean accuracy (%) of test nodes as input graph contains varying number of connected components. For each graph dataset, we consider three scenarios: all components, all components except isolated nodes (i.n.), and, only the largest component. Listed in brackets next to each dataset name is the total number of components in each graph.

| Model | Cora (78) | | | Citeseer (438) | | | Amazon Computers (314) | | | Amazon Photo (136) | | |
|---|---|---|---|---|---|---|---|---|---|---|---|---|
| | all | no I.N. | largest | all | no I.N. | largest | all | no I.N. | largest | all | no I.N. | largest |
| GCN | 79.2 | - | 81.3 | 6.61 | 66.7 | 71.4 | 2.9 | 82.3 | 82.5 | 4.4 | 90.5 | 91.0 |
| GAT | 80.8 | - | 82.3 | 68.2 | 68.4 | 71.2 | 80.1 | 79.2 | 76.1 | 87.9 | 80.3 | 87.4 |
| MoNet | 80.3 | - | 82.0 | 67.7 | 68.1 | 71.1 | 83.2 | 84.5 | 84.5 | 90.2 | 91.2 | 91.3 |
| GS-mean | 79.0 | - | 80.6 | 67.6 | 67.8 | 71.6 | 81.0 | 82.3 | 83.4 | 89.9 | 91.1 | 91.4 |
| GS-maxpool | 75.9 | - | 77.3 | 63.6 | 64.3 | 67.4 | - | - | - | 89.0 | 90.1 | 90.1 |
| GS-meanpool | 76.9 | - | 78.8 | 64.4 | 65.4 | 68.5 | 78.1 | 80.2 | 80.3 | 88.8 | 90.4 | 90.6 |
| LabelProp-NL | 68.9 | - | 74.3 | 47.4 | 47.5 | 66.4 | 72.5 | 74.0 | 75.7 | 81.6 | 83.8 | 81.0 |
| LabelProp | 69.3 | - | 75.1 | 47.3 | 47.6 | 67.8 | 70.0 | 68.6 | 67.0 | 78.4 | 75.4 | 67.0 |
| Avg. GNN | 78.7 | - | 80.4 | 56.3 | 66.8 | 70.2 | 65.1 | 81.7 | 81.4 | 75.0 | 88.9 | 90.3 |
| Avg. LabelProp | 69.2 | - | 74.7 | 47.4 | 47.6 | 67.1 | 71.3 | 71.3 | 71.4 | 80.0 | 79.6 | 74.0 |

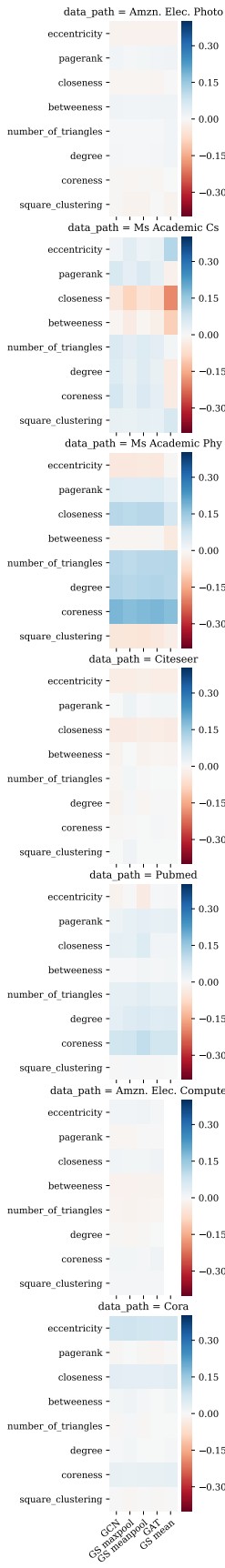

Figure 6: An expanded list of all correlations datasets and models. Accuracy and topological features show weak correlation (Pearsons r) even across these. However as the number of points decreases with increasing granularity there is lower confidence ($p$) in these results.

# E    Noise Algorithm

The algorithms perturbs an edge with probability $\mathcal{P}$, and thresholds the resulting edges with respect to a parameter $t$. By varying the threshold for an edge to exist, we created different perturbed versions for each dataset. Note that a smaller $\mathcal{P}$ means a higher probability for edges in the original graph to be retained, whereas a higher $t$ means a smaller number of new edges are added to the graph.(For the following algorithm $p = \mathcal{P}$)

---

**Algorithm 1:** perturbation and threshold method

---

**Input:** Adjacency Matrix $A$, number of nodes: $n$, Threshold: $t$, an edge probability: $p$
**Result:** A adjacency matrix $A^{pt}$
mx = $max(A)$;
mn = $min(A)$;
$A^{mask}$ = Erdos-Renyi(n, p).adjacencyMatrix;
$A^{rnm} = A^{mask} \odot \mathcal{N}(0,1)[n,n]$
rnmn = $min(A^{rnm})$;
rnmx = $max(A^{rnm})$;
$A^{scaled} = $ mn $+ (A^{rnm} - $ rnmn$)/($rnmx $- $ rnmn$) * ($mx $- $ mn$)$
**for** $i,j : i,j \in \mathbb{I}^+, i,j < n$ **do**
    **if** $A^{mask}_{i,j} == 0$ **then**
        $A'_{i,j} = A_{i,j}$;
    **else**
        $A'_{i,j} = A^{scaled}_{i,j}$;
    **end**
**end**
$A^{p,t} = A' \odot heaviside(A' - t))$

---

# F    PCA Experimental setup

4 types of constructions were made. The exact configuration depended on whether the high or low variance attributes were discarded and whether the discarded edges were used to augment the dataset:

1. high Inf: kept; Low inf: discarded [top left]
2. high Inf: kept; Low inf: edges [top right]
3. high Inf: discarded; Low inf: kept [bottom left]
4. high Inf: edges; Low inf: discarded [bottom right]

To prevent "noisy" edges from being added only the most similar $60 * density_G$ percentile pairs were chosen. While this ensure the same order of edges are added, the exact number is subject to hyperparamter tuning.

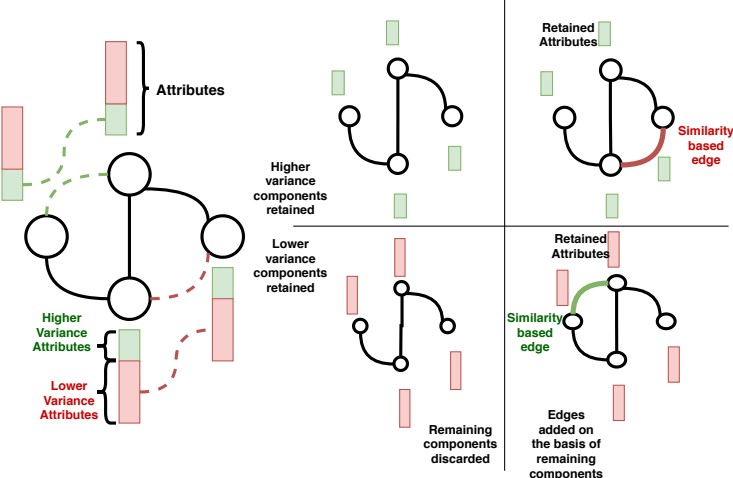

Figure 7: Graphs constructed based on 4 possible cases arise depending on whether higher or lower variance features are retained and whether the remainder are discarded out of hand or used to create edges before discarding. The original graph on the left with attribute vectors next to each node. Dashed lines between nodes depict possible edges and between parts of the attribute vector depict high similarity. Graphs on the right show the constructed graphs.

