# OpenReview forum: "The Surprising Behavior Of Graph Neural Networks"
_ICLR.cc/2020/Conference — Reject_

### Official Review · AnonReviewer1 · 2019-10-21
**Official Blind Review #1**

**Rating:** 6

**Review:**

This paper analyzes the properties of graph neural networks (GNNs). It shows that several hypothesis that one might intuitively make about the behavior of GNNs, do not actually hold. In fact, some observations are even contradictory, indicating that GNNs' performance is not robust, and care needs to be taken when using them. In particular, the authors analyze what happens when topology is altered by computing correlations between topology metrics and accuracy, dropping connections, adding extraneous connections, etc. The authors examine performance is a function of graph connectedness, and find a weak correlation.

Some concerns:
a) The part about attributes and topology is not very clear to me. What are the attributes? What's "decoupling by shuffling"?
b) Figures could benefit from a brief "so what" explanation in the caption.
c) While the work is important because GNNs are a rising trend, it is a bit disappointing that there is no discussion of "how do we fix GNNs" and "what's next".
d) Topology is one important feature of graphs, but could it be examined in terms of what kinds of edges are added based on learned similarity metrics, etc? Learning the adjacency matrix is one important step in GNN methods and it would be useful to examine robustness to different ways of learning that matrix.

**Experience Assessment:**

I have read many papers in this area.

**Review Assessment: Checking Correctness Of Derivations And Theory:**

I assessed the sensibility of the derivations and theory.

**Review Assessment: Checking Correctness Of Experiments:**

I assessed the sensibility of the experiments.

**Review Assessment: Thoroughness In Paper Reading:**

I read the paper thoroughly.

---

> ### Author Response · Authors · 2019-11-15
> **Incorporating feedback into this version**
>
> Thank you for your discerning review. We hope to have addressed most of your concerns:
> -We have added an explanation of different aspects of a complex networks in the dataset’s section
> -We have incorporated more comprehensive figure captions
> -The discussion and conclusion sections present the improvement our experiments form the basis of and go over “what’s next” i.e. future work.
> -One of the limitations of the work is the similarity function/or distance metric used. We have provided more details about the choices of function that were used such as the reasons behind selecting cosine metrics, and highlighted the limitation in our discussion.

---

### Official Review · AnonReviewer2 · 2019-10-23
**Official Blind Review #2**

**Rating:** 3

**Review:**

The paper presents four experimental set-ups to get a better understanding how the performance of Graph Neural Networks (GNN) depend on topological and/or nodal information.

As the paper is not really in my research area, I would have liked the paper to be a bit more self-contained, but the writing of the paper is generally clear.

The contributions of the paper are mainly experimental and show different "surprising" aspects about GNNs. I found the experimental results to be interesting, especially those presented in Section 5 about decoupling attribute and topological information. However, I think the presentation may be improved and more information (e.g. about how the experiments were conducted or more plots) should be reported in the main paper or the appendix.

Regarding the experiments, the authors made choices about which subsets of datasets (Table 2) or which subsets of measures (Table 1) to use. It would be nice to explain how those choices are made and if they are well-justified. Otherwise, it feels that the presented experimental results have been hand-picked.

Regarding Section 3, I guess that Figure 1 presents averaged results over datasets or over models. I think it would be important to share the results per datasets or per models, as averaging may hide some key aspects.

On page 3, the third paragraph could be illustrated with some plots. Besides, the its first sentence seems to contradict the last sentence of the paragraph before Section 4. Moreover, the text in the fourth paragraph doesn't seem to fit well Fig. 2b for Amazon computers. Am I misreading this plot?

Regarding Section 4, I think it would have been interesting to report the metrics about the topological features for the perturbed graphs. Are they very different from those for the orignal graphs?

Regarding Section 6, the sentence "In the previous sections, we have demonstrated that GNNs .. are robust to perturbations to it" on page 7 seems to contradict the conclusion of Section 4.
I found the experiments in this section to be less convincing. The edge addition technique seems to be ad-hoc and it is not clear why it would improve the performance or not. For instance, how was the value 6°*density-G chosen?

Overall, although I found some of the experimental results very intriguing, I think the paper may not be ready for publication in its current state.

**Experience Assessment:**

I do not know much about this area.

**Review Assessment: Checking Correctness Of Derivations And Theory:**

N/A

**Review Assessment: Checking Correctness Of Experiments:**

I assessed the sensibility of the experiments.

**Review Assessment: Thoroughness In Paper Reading:**

I read the paper at least twice and used my best judgement in assessing the paper.

---

> ### Author Response · Authors · 2019-11-15
> **Incorporating feedback into the paper**
>
> Thank you for your detailed comments. They were most helpful and we’ve incorporated all of them into our draft.
>
> We have explained the reasons behind the selection of certain features and datasets more clearly:
> - Certain features can only be calculated at a graph level and others at node level (e.g. betweenness is calculated for each node); and
> - We’ve selected datasets which span different types of connectivity scenarios (e.g. well connected with no isolated nodes vs. several disconnected components).
>
> We have augmented the text with appendices that show:
> - Non-averaged figures for section 3; and
> - Topological Features of the perturbed datasets.
>
> We have also significantly redrafted the paper to increase clarity:
> - The method section (section 6)
> - In section 3, we redrafted this section to clarify the experimental results to get rid of any inconsistencies in the text. We have  further supported the results realting isolated nodes by drawing the readers attention to table 5 in the appendix - which provides a more detailed description of the results.
> - In section 6, we have redrafted with a strong emphasis on explaining our assumptions. For instance, while somewhat a hyperparameter, the percentile threshold of 60*density(G) was chosen only to retain the most similar pairs and minimizing noise in the augmented edge set. This is now reflected in the text.

---

### Official Review · AnonReviewer3 · 2019-10-23
**Official Blind Review #3**

**Rating:** 1

**Review:**

This work empirically  study the behavior of Graph Neural Networks in various topological contexts. Four sets of experiments are provided follow the setting in [1].
Pros:
1. The research problem studied in the paper is important.
2. Authors conduct extensive experiments on multiple dataset.
Cons:
1. The paper lacks formal justifications on the raised claims. They look intuitive and post-justified by experiments and not by rigorous arguments.
2. All the experiments are conducted based on the convolutional graph neural network based methods. It is suggested to evaluate on the other types of graph neural network, e.g. Recurrent Graph Neural Networks, Graph Autoencoder.
3. The writing needs to be significantly improved.

[1] Shchur et al. "Pitfalls of Graph Neural Network Evaluation",  arXiv preprint arXiv:1811.05868 (2018)

**Experience Assessment:**

I have published one or two papers in this area.

**Review Assessment: Checking Correctness Of Derivations And Theory:**

I carefully checked the derivations and theory.

**Review Assessment: Checking Correctness Of Experiments:**

I carefully checked the experiments.

**Review Assessment: Thoroughness In Paper Reading:**

I read the paper thoroughly.

---

> ### Author Response · Authors · 2019-11-15
> **Incorporating feedback into the paper**
>
> We thank the reviewer for their insight and comments.
>
> The reviewer astutely points out that the paper looks primarily at convolutional and an attention based model (according to the taxonomy in https://arxiv.org/pdf/1901.00596.pdf). We’ve scoped the paper to explicitly mention this both by changing the title and reiterating it within the methods section. Expanding the paper to include additional models is in our timeline for future exploration.
> We also agree that the paper is an empirical one; and one of its primary goals is to emphasize the lack of a single theory to explain this behaviour of convolutional graph neural networks. We’ve more explicitly stated its empirical nature and expanded the discussion to better describe the possible causes behind the experiments.
> In relation to the issue of post-justification, this paper’s main purpose is to explore unexpected behaviours for GNN’s and as such is less an explication of an overall theory, and more a first step in better mapping and analysing these behaviours. The appearance of results being intuitive stems from the paper’s primary purpose in contrasting this intuitive understanding with experimental proof.
> Finally we’ve significantly revised the writing and hope this will improve clarity of content and highlight latent points.

---

### Author Response · Authors · 2019-11-15
**Overall improvements in this version**

We thank everyone for their insight and reviews. They allowed to further our conclusion and ideas.

Our paper is an empirical study that attempts to a highlight a gap in our understanding as researchers of the behavior of convolutional graph neural networks.  It does not present a consolidated theoretical model of their behavior. We explore GNN’s ability to operate across domains and scratch the surface of GNN’s claim to utilize the topology of Real-world complex networks.

These experiments serve to highlight integral aspects of the impact of a network’s topology and the distribution of node attributes to a model’s performance. They are not comprehensive and representative all possible scenarios, but rather highlight important places where our intuitive understanding differs from experimental behavior. Scenarios which occur in the real world and where the application of these models may have unforeseen consequences.

We have improved the explanations, clarifying several procedures and providing the reasoning behind our choices. We have also augmented the appendices with extra diagrams and data where possible, and would be happy to share the datasets and other material after the paper is deanonymized.

---

### Decision · Program_Chairs · 2019-12-19

**Decision:**

Reject

**Comment:**

The paper empirically investigates the behaviour of graph neural networks, as a function of topology, structural noise, and coupling between nodal attributes and structure. While the paper is interesting, reviewers in general felt that the presentation lacked clarity and aspects of the experiments were hard to interpret. The authors are encouraged to continue with this work, accounting for reviewer comments in subsequent versions.